# Personalized Approach in Treatment of Neovascular Age-Related Macular Degeneration

**DOI:** 10.3390/jpm12091456

**Published:** 2022-09-05

**Authors:** Radina Kirkova, Snezhana Murgova, Vidin Kirkov, Ivan Tanev

**Affiliations:** 1Department of Ophthalmology, ENT and Maxillofacial Surgery, Medical University—Pleven, 5800 Pleven, Bulgaria; 2Department of Ophthalmology, IRCCS Humanitas Research Hospital, 20089 Rozzano, Italy; 3Department of Ophthalmology, University Hospital Dr. Georgi Stranski, 5800 Pleven, Bulgaria; 4Department of Health Policy and Management, Faculty of Public Health “Prof. Dr. Tzekomir Vodenicharov”, Medical University of Sofia, 1000 Sofia, Bulgaria; 5Eye Clinic “Zrenie”, 1000 Sofia, Bulgaria

**Keywords:** OCT-A, neovascularization, AMD, anti-VEGF, vascular remodeling, personalized, approach

## Abstract

Background: Age-related macular degeneration (AMD) is a progressive, degenerative disease of the central retina. AMD is subdivided into “dry” (atrophic), “wet” (exudative), and neovascular (nAMD) forms. In recent years, the concepts about nAMD changed with the development of optical coherence tomography–angiography (OCT-A) and intravitreal anti-VEGF treatment. The aim of this study was to define the morphologic type of the neovascular membrane (NVM) before treatment with OCT-A and to register vascular remodeling after treatment with anti-VEGF. We also analyzed the relationship between NVM and visual acuity. Methods: The study was retrospective and included 119 patients with newly diagnosed, treatment-naïve nAMD. All the patients underwent full ophthalmic examination and also fluoresceine angiography and optical coherence tomography–angiography (OCT-A). Results: Based on the collected data, we found repetitive regularities. Conclusion: The analysis of our results could be used as prognostic markers for the evolution of the disease and as a basis for new treatment strategies, depending on the naïve NVM morphologic type.

## 1. Introduction

AMD is a progressive, degenerative disease, involving the central retina. It was first described by Hutchinson in 1874 as a “symmetric chorioretinal disease in old people”. Many years later, the term “age-related maculopathy” was introduced. Nowadays, the end stage of the disease is called AMD. The disease comprises 8.7% of the reasons for blindness worldwide [1,2,3,4] and it occupies third place, after cataracts and glaucoma [5]. According to the World Health Organization (WHO), in 2040, patients with AMD will exceed 288 million. AMD is subdivided into “dry” (atrophic), “wet” (exudative), and neovascular. All the forms result in irreversible vision loss and heavy impairment of the quality of life of the patients [5]. Before the introduction of anti-VEGF for ophthalmic purposes in 2006, nAMD was considered the worst because of the development of new vessels, which leads to exudation and bleeding in and under the neurosensory retina, resulting in fast vision loss. For the diagnosis of nAMD, we need ophthalmoscopy, but also highly specialized tests such as fluorescein angiography (FA), indocyanine green angiography (ICG), optical coherence tomography (OCT), and OCT-angiography (OCT-A). OCT and OCT-A were developed as devices for early diagnosis and registration of the stage and the size of the lesions in patients with AMD [6,7,8]. The use of OCT-A in everyday clinical practice leads us to the necessity of introducing new diagnostic parameters and classifications because of its unique features and specific relationships between imaging and histology, much more complex than those of the other imaging techniques available in the ophthalmology [9].

## 2. Aim and Scope

The aim of this study was to evaluate the abilities of OCT-A in the diagnosis and in the follow-up of the results after treatment with intravitreal anti-VEGF in patients with nAMD.

Aim 1: Defining the type of NVM before treatment (naïve), according to OCT-A images.

Aim 2: Assessment of the changes in the type of NVM after treatment with intravitreal anti-VEGF (vascular remodeling).

Aim 3: Assessment of the relationship between visual acuity (VA) before treatment and VA after treatment.

Aim 4: Assessment of the relationship between the type of NVM (before and after treatment) and VA (before and after treatment).

## 3. Materials and Methods

All materials were collected in the following clinics:- Eye clinic “Zrenie”—Sofia, Bulgaria- Eye clinic in University Hospital for Active Treatment “Georgi Stranski”—Pleven

Our study includes all the patients with newly diagnosed, treatment-naïve nAMD. The study was retrospective and included 119 patients (Table 1).

All the data were collected in the period November 2018–December 2021. The algorithm for diagnosis and evaluation of patients was clearly defined and equal for all patients with newly diagnosed, treatment-naïve nAMD. They underwent full ophthalmic examination:- Full anamnesis—family history, risk factors, allergies, systemic diseases- Visual acuity for distance and near through chart with Snellen optotipes. If the patient could not see optotipes, we examined their ability to count fingers and register the moving of a hand. In patients with very low visual function, we assessed their ability to detect light—perception and proection and if they can define the color of the light (“red” or “green”).- Tonometry (“air-puff” tonometer)- Biomicroscopy—assessment of orbit, eyelids, lid margin, conjunctiva, cornea, iris, pupillary reactions, anterior chamber, lens (also LOCS grading), or IOL- Indirect ophthalmoscopy—assessment of vitreous body, optic nerve, retina (ophthalmoscopic signs of exudative activity or drusen)

All the patients underwent FA (“gold standard”), structural OCT and OCT-A. Main scientific interest of this study is to assess vascular remodeling after anti-VEGF treatment through OCT-A. We used OCT-A machine Nidek RS-3000 Advance 2 (software version NAVIS-EX 1.8.0).

After performing all the tests described above and confirming the diagnosis, we proceed to a detailed evaluation of OCT-A images in their three different depths. In our research we included only eligible OCT-A scans with good quality of the image. All the scans were assessed by one experienced examinator. The type of neovascular membrane was determined by this examinator, according to the classification described above and presented by Coscas et al. [10]. The cohort of 119 patients was divided into several groups, based on the morphological appearance of the neovascular membrane from the OCT-A image.

Patients with:NV-membrane type “Sea fan”—with eccentric feeding vessel, massive trunks with thin capillary ramifications (Figure 1A)NV-membrane type “Medusa”—has a massive feeding vessel with centrifugal vascular trunks with thin capillaries (Figure 1B)NV-membrane type “Dead tree”—has a massive main trunk and ramifications varying in size and caliber (Figure 1C)NV-membrane type “Lace”—highly anastomotic vascular network, without main vessel (Figure 1D)NV-membrane type “Filaments”—composed of many intertwining, filamentous vessels (Figure 1E)nondetermined type of NV-membrane—when the OCT-A appearance does not correspond to any of the above types

Of the approved anti-VEGF drugs in Bulgaria, Eylea (Bayer) is the most affordable, as it is reimbursed by the National Health Insurance. Our aim was not to compare the therapeutic effect of the different anti-VEGF drugs, available on the market, but to demonstrate the potential of the OCT-A in diagnosis and personalization of the therapeutic protocol, resulting in better structural and visual outcome. Therefore, only patients treated with the same drug were included in the study.

The anti-VEGF administration procedure is performed in an operating room under sterile conditions. Seven days before the intravitreal injection, the patient puts antibiotic drops for prophylaxis. After the placement of blepharostat, we perform anesthesia with Alcaine drops and double exposure to Betadine solution for 2 min, intrapalpebral. Using a caliper and depending on the patient’s phakic status, Eylea (Bayer) is injected intravitreally into the lower nasal quadrant of the eyeball. The patient is bandaged until the next day. Control examinations, including assessment of visual acuity, tonometry, biomicroscopy, ophthalmoscopy, OCT, and OCT angiography, were performed on day 25 after the injection. Regarding admission to the current study of the therapeutic protocol, the next injection should be given after 1 month from the start, with the patient facing three “loading” doses one month apart. After each intravitreal injection, special attention must be paid to visual acuity as a functional measure of the effect of the treatment. In OCT-A images, a change in the morphological shape of the membrane is determined. Signs of progression are monitored—persistence, increase, or regression.

## 4. Statistical Analysis

In order to make the follow-up easier and more representative, the collected data were summarized and entered in a tabular form in Microsoft Excel. Each individual patient was introduced on a new line, and the indicators of morphological appearance of the naïve NV-membrane (NVM), morphological appearance of the NVM after three intravitreal applications of anti-VEGF, assessment of signs of progression and visual acuity before and after therapy were introduced in columns. Statistical analysis was performed using the software package SPSS, 13.0 (SPSS Inc., Chicago, IL, USA). All values with *p* < 0.05 were considered statistically significant.

## 5. Results

According to Aim 1, the individual types of NV membranes were defined before starting therapy according to their OCT-A characteristics. The largest number was of the “sea fan” type (40 patients), followed by “lace” (25 patients), “filaments” (23 patients), and “Medusa” (19 patients). The obtained results are represented graphically in Figure 2.

Although age is the major risk factor for the development and progression of the disease, no statistical relationship has been established between the age of the patients and the naïve form of the neovascular membrane. No relationship was found between the sex of the patients and the shape of the “naïve” neovascular membrane.

Aim 2: to determine whether there is a relationship between the initial appearance (naïve neovascular membrane) and its change after three applications of anti-VEGF intravitreal in a month. Statistical analysis proved a relationship between the morphologic form before therapy and that after for the whole cohort of 119 patients: Spearman’s rho: *ρ* = 0.61137, *p* = 0, *n* = 119. This relationship is graphical represented in Figure 3.

For a more detailed and in-depth analysis, naive neovascular membranes were further subdivided according to the “maturity” of the vessels. This way, we seek for connection between the histological structure of the membrane (the components that make up its vessel wall) and the subsequent remodeling after therapy. The subdivision was performed taking into account the OCT-A image and caliber of the vessels forming the membrane and according to the theory of retinal angiogenesis of De Almodovar et al. [11].

Thus, two groups were formed: “differentiated” and “undifferentiated” membranes. The “differentiated” include “sea fan”, “Medusa”, “dead tree”. To the group of “undifferentiated” we refer—“lace”, “filaments”, “undetermined”. For the group of differentiated (*n* = 62), the Spearman’s rho test was applied again, which confirmed the statistical correlation between the three morphological species: Spearman’s rho: *ρ* = 0.32385, *p* = 0.01024, *n* = 62. A chi-square test was also applied—39.6406, *p* = < 0.00001, which confirmed once again the statistical relationship (the results are visible in Table 2).

The study found that neovascular membranes of the “sea fan” type most often after therapy became “Medusa” (47%) or “dead tree” (33%). Only 20% did not change after therapy. “Medusa” type after therapy in 55.6% passed into a “dead tree”, and in 44.4% did not change. In 75% of cases, the “dead tree” retained its morphological appearance after therapy. It is a final, maximally differentiated form (histologically) of a neovascular membrane, that does not change over time. The results are represented on Figure 4.

For the group of “undifferentiated”, immature neovascular membranes, which include “lace”, “filaments” and “nondetermined”, Spearman’s rho and chi-square test were applied again. However, they did not find a correlation between the different forms before and after therapy. Although in 64% of cases the naive neovascular membrane type “lace” does not change after therapy, the statistical data for the three types do not allow defining a relationship between naïve forms and the morphologic type of the membrane posttreatment.

According to requirement of Aim 3, we had to check if there is a link between visual acuity before and visual acuity after therapy. For the entire cohort of 119 patients and then for the two additionally subdivided groups (of “differentiated” and “undifferentiated” neovascular membranes), the statistical analysis showed an extremely strong relationship:

“The higher the initial visual acuity, the higher it is after therapy”.

In searching for the results of the Aim 4, an attempt was made to establish a connection between the individual morphological variants of naïve neovascular membranes and visual acuity. However, this has not been proven statistically. i.e., the data from the study did not prove a relationship (either positive or negative) of any naïve morphological type of membrane with visual acuity before and after therapy.

After the first three “loading” doses of anti-VEGF and the registered on OCT-A vascular remodeling, a statistical relationship was established between the morphological variant after therapy and visual acuity for each group

For the group of “differentiated” neovascular membranes (“sea fan”, “Medusa”, “dead tree”) the dependence is negative:

Spearman’s rho: *ρ* = −0.29387, *p* = 0.02043, *n* = 62.

This means that the less differentiated the membrane (for example, “sea fan”), the lower the visual acuity.

For the group of undifferentiated membranes, again we have a statistically significant result, but it has a positive sign:

Spearman’s rho: *ρ* = 0.30498, *p* = 0.02228, *n* = 56, i.e., the more differentiated the membrane (e.g., “filaments”), the higher the visual acuity.

## 6. Discussion

OCT-A was first demonstrated in 1997 by Chinn et al. [12] but they did not find applications in clinical practice—the quality of images was low due to a lack of technological capabilities. In the next decade, the technology developed and in 2006 Makita et al. presented the first OCT-A. It is based on a new method for analysis of high-resolution imaging techniques without the need to introduce contrast. OCT-A is based on the concept that in static eye tissues, the only moving structures are the formed elements of the blood as they pass through the vessels [13]. In clinical practice, OCT-A allows us to represent classical and occult neovascular membranes and gives detailed information about the choriocapillaris. OCT-A presents the retinal blood flow divided into two plexuses [14]: superficial and deep (formed by the complex internal/external plexus by morphological data). OCT-A allows us to study and to evaluate separately the two vascular plexuses and their demonstration in vivo. In fluorescein angiography (FA), the two plexuses overlap and cannot be evaluated separately, as the resulting images are cumulative. The possibility of visualization of the two plexuses separately is one of the main advantages of OCT-A. OCT-A demonstrates precisely the intravascular blood flow without contrast. When analyzing the image from the OCT-A, attention should be paid to the localization and to the depth of the scan with characteristics such as reflectivity, flow, morphology, and architectonic. We evaluate the four depths of scanning [15]: superficial plexus, deep plexus, RPE/Bruch membrane complex, and choroidal. The correct interpretation of the OCT-angiography image requires consideration of “en face” and transversal scans at each of the specified levels so that the clinical correlation can be maximally correct. For each of the four depths of scanning should be determined: flow and decorrelation, vascular morphology and architectonics, texture—all up to one new indicator characterizing OCT-A and differing it from FA [16]. There is a well-known classification of neovascular membrane types (NVMs) according to the findings in the FA images. Type I NV (occult) is the most common: it starts from the choroid and reaches the Bruch membrane and retinal pigment epithelium (RPE). Type II NV is located subretinally. Type III—intraretinal (retinal angiomatous proliferations—RAP). Although the FA has been the gold standard in the diagnosis of neovascular AMD for more than 50 years, the classifications used for it are not applicable to OCT-A. Palejwala et al. was the first to report the application of OCT-A for early detection of type I NV. El Ameen et al. described type II NV using OCT-A. Other studies reported the administration of OCT-A in the detection of type III NV (RAP). The introduction of the wide use of OCT-A into daily clinical practice led to the necessity of new diagnostic parameters and new classifications. Coscas et al. in 2015 examined with OCT-A 80 eyes with exudative AMD and determined different morphological variants of NV. According to Coscas et al. and other researchers [15] of recent years, based on the images from OCT-A, NV membranes morphologically are defined as “seaf an”, “Medusa”, “dead tree”, “lace”, and “filaments” [17,18].

The results of ours study proved a link between the “naïve” morphological forms of neovascular membranes and defined a dependency on switching one naïve form to another after therapy—the so-called vascular remodeling. For more accurate and thorough interpretation and taking into account De Almodovar et al.’s theory of vascular genesis, the “naïve” neovascular membranes were further subdivided into groups depending on the caliber of the vessels building them (the histological structure, respectively). The larger the caliber of the vessels building the NV membrane, the “more mature” in histological terms they are. The study found that “naïve” NV membranes from the group “differentiated” (“sea fan”, “Medusa”, “dead tree”) tend to pass into each other after therapy – graphically represented on Figure 5.

Another important regularity is the relationship of histological differentiation with visual acuity after therapy. It was found that the more differentiated a membrane (for example, “dead tree”), the higher is the visual acuity after treatment. For less differentiated membranes, no relationship for vessel remodeling has been proven. The relationship of the remodeled type of neovascular membrane (group “non-differentiated”: “lace”, “filaments”, “nondetermined”) was established after therapy with visual acuity. The more highly differentiated, and respectively histologically “mature” the type of the membrane, the higher the visual acuity. Given the tendency in highly differentiated naïve forms of neovascular membranes to pass into each other and their relationship to visual acuity after therapy, it can be concluded that when establishing a naïve form “sea fan”, “Medusa”, “dead tree”, it is possible to predict the evolution of the disease and visual acuity after therapy.

## 7. Conclusions

Age-related macular degeneration increases its social importance because of the aging pyramid of the population. In its late stages, the disease is highly debilitating, disrupting the quality of life of patients and their ability to take full care of themselves. The neovascular form of AMD affects many layers of the outer and inner retina, the RPE, and the choroid. Multimodal imaging techniques (FA, Indocyanine angiography, OCT, OCT-angiography) are used in the diagnosis and follow-up of patients. Due to the possibilities for segment separation and analysis of microcirculation at different levels, OCT-A has become an integral part of the diagnosis and follow-up of patients with AMD. OCT-A improves the identification capabilities and complements structural OCT, FA, and indocyanine angiography. It is non-invasive, fast, and brings the ophthalmologist detailed information. The detected patterns for vascular wall remodeling could serve as prognostic markers for the evolution of the disease and to become a basis for developing new therapeutic regimens and individualizing the approach to the “naïve” type of NV membrane at the time of diagnosis of neovascular AMD.

## Figures and Tables

**Figure 1 jpm-12-01456-f001:**
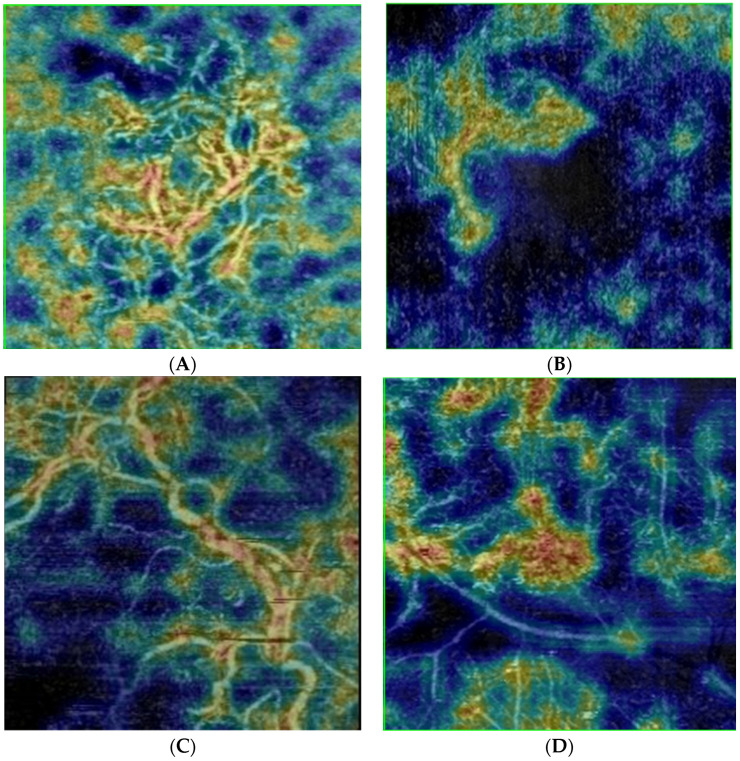
ОСТ-angiography, Eye Clinic ”Zrenie“; Nidek OCT-A RS-3000) Advance 2. (**A**) NV type “Sea fan”; (**B**)NV type “Medusa”; (**C**)NV type “Dead tree”; (**D**). NV type “Lace”; (**E**) NV type “Filaments”.

**Figure 2 jpm-12-01456-f002:**
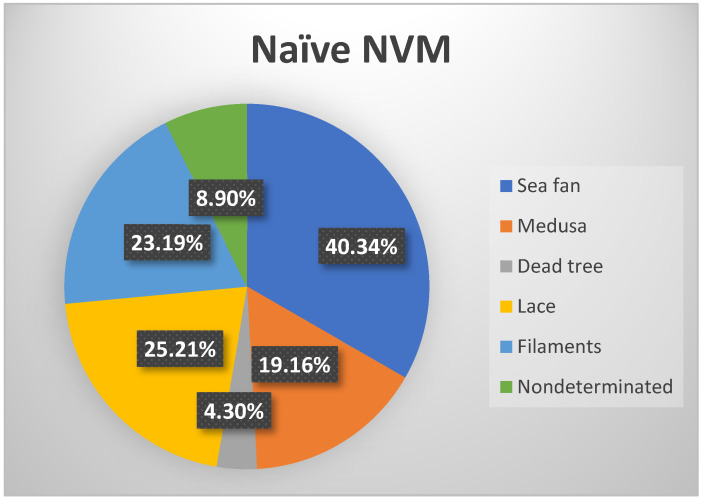
Percentage of naïve morphological type of NVM.

**Figure 3 jpm-12-01456-f003:**
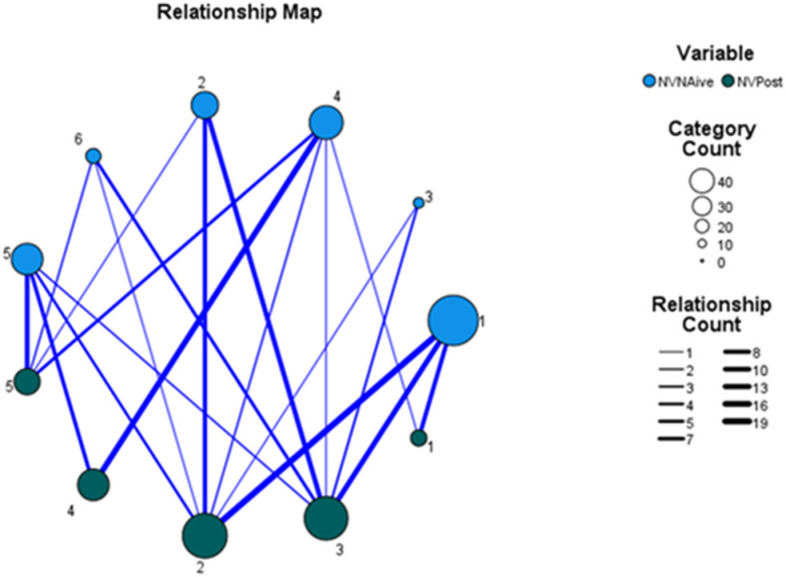
Relationship map: between the naïve type of NV membrane and the type after therapy: The thickness of the line is directly proportional to the strength of the connection. Legend: NV naïve—“naive” form of a neovascular membrane, before therapy; NV post—a form of the neovascular membrane after three intravitreal anti-VEGF applications with an interval one month. 1—Sea fan; 2—Medusa; 3—Dead tree; 4—Lace; 5—filaments; 6—undetermined.

**Figure 4 jpm-12-01456-f004:**
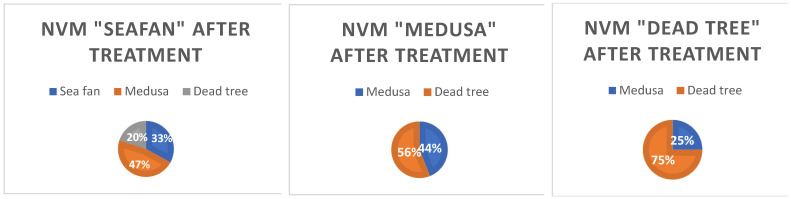
Remodeling NV type “sea fan”, “Medusa”, “dead tree” after therapy.

**Figure 5 jpm-12-01456-f005:**
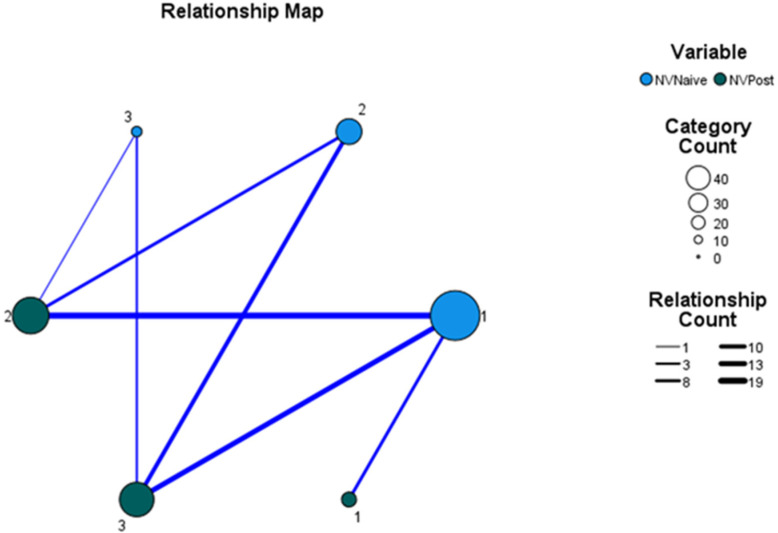
Relationship map—between the naïve type of NV membrane and the type after therapy—group “differentiated”. The thickness of the line is directly proportional to the strength of the connection. Legend: NV naïve—“naïve” form of a neovascular membrane, before therapy; NV post—A form of the nonvascular membrane after three. intravitreal anti-VEGF applications with a month interval between them. 1—Sea fan; 2—Medusa; 3—Dead tree.

**Table 1 jpm-12-01456-t001:** Demographic characteristics of the followed cohort.

Age	Males	Females	Total
Mean age	75.75	75.80	75.78
SD	9.65	9.46	9.61
Median	77	77	77
Range	53	66	119

**Table 2 jpm-12-01456-t002:** Results of the chi-square test.

Results			
	NV Naïve	NV after Treatment	All Rows
Sea fan	40 (24.00) [10.67]	8 (24.00) [10.67]	48
Medusa	18 (23.00) [1.09]	28 (23.00) [1.09]	46
Dead tree	4 (15.00) [8.07]	26 (15.00) [8.07]	30
**All columns**	62	62	**124** (Total)

## Data Availability

All the data and materials are saved in the archives of Eye Clinic “Zrenie”: Sofia, Bulgaria. The information about patient’s demographic characteristics, visual acuity (as a comment in each patient’s folder) and OCT-A scans are saved in Nidek RS 3000 Advance (software version NAVIS-EX 1.8.0.) in Eye Clinic “Zrenie”, Sofia, Bulgaria.

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
