# Peer review of "Personalized Approach in Treatment of Neovascular Age-Related Macular Degeneration"

_jpm, 2022, doi:10.3390/jpm12091456_

Round 1

Reviewer 1 Report

The Authors Radina Kirkova et.al in the article entitled ‘Personalized approach in treatment of neovascular age-related macular degeneration’ have a non-invasive, novel strategy for detecting patterns of vascular wall remodeling as potential prognostic markers in the diagnosis of neovascular AMD. The authors have provided a thorough introduction. The authors have conducted the study well by applying right statistical tools via SPSS to evaluate the data.

It is appreciable that the authors have highlighted the Aims and scope of the study and have presented the data accordingly.

Minor Queries:

1.     To assess the risk factors, is the occupation status and living condition update of the cohorts recorded?

2.     Line# 48:  What does OCT-Q mean? Or is it a typo (OCT-A)  

3.     Are there any representative images from the structural OCT imaging to observe the retinal thickness?

Author Response

Dear Reviewer,

Thank you for your time, patience and your precious comments.

I would want to answer to your queries point by point:

Question №1.     To assess the risk factors, is the occupation status and living condition update of the cohorts recorded?

All the patients, included in our study had no specific risks of exposure on blue light or excessive exposure to UV light. The cohort of 119 patients is from an urban area, no one with agriculture background. All of them are on a typical for the region Mediterranean diet.

Question №2.     Line# 48:  What does OCT-Q mean? Or is it a typo (OCT-A) 

Yes, it is a typo – OCT-A

Question №3.     Are there any representative images from the structural OCT imaging to observe the retinal thickness?

Yes, we have. We have done also structural OCT scans to all the patients (modalities: “line”, “cross”, “macula” [retinal thickness]. We don’t include the data in this article, because we wanted to focus and put the accent on the remodeling of neovascular membranes and the features of OCT-A in the follow-up of the patients.

At this point our Clinic is in summer holidays and I can’t send you a scan with central retinal thickness, but I would want to demonstrate you the relationship between the OCT-A and the structural OCT in one of the patients with representative images, included in the study (you ca find it as an attached file)

I can provide you other images next week.

I hope my responses are comprehensive and will satisfy you.

For any queries, I remain on your disposal

Reviewer 2 Report

In the current study, Kirkova et al described the patterns of vascular remodeling after three anti-VEGF treatments in 119 treatment-naïve nAMD patients. Vascular remodeling is a process whereby angiogenesis is repeatedly produced and dissipated according to repeated anti-VEGF treatments over a longterm. Thus, the interval of monitoring patients in this study is relatively short. This reviewer highly recommends a comparison of the results at least 1 year after the first treatment. The following are additional important issues that should be addressed by the authors.

1.       The accuracy of NV-membrane type classification is the most fundamental part of this study. However, only one examiner was responsible for this classification, whereas in most studies at least two examiners conduct classification.

2.       In Figure 1, the representative images of each type are shown. However, the quality of the images are poor, especially Figure 1D.

3.       In Table 1, all the mean age of males, females, and total patients are the same, 76 years old. Is this a true representation?

4.       There are several sentences that need to be referenced.

5.       There are several typos. Please edit the manuscript carefully.

Line 18: optic

Line 49: OCT-Q

Line 82: standart

Line 131: HB-membrane

Line 152: p=0

Author Response

Dear Reviewer,

Thank you for your time, patience and your precious comments.

I would want to answer to your queries point by point:

Anti-VEGF therapy seems to induce a quantitative regression with a variable decrease in size and vessel density of the neovascular membrane, which has been described from 2 to 9.5 weeks after treatment[i], [ii]. A pruning of smaller vessels occurred 24 hours after injection, increased and reached a maximum of flow regression between 6 days and 12 days, followed by a reproliferation (reopening or new sprouting of the vessels) 20 days to 50 days later[iii]. Moreover, in a qualitative approach, Spaide depicted vascular “abnormalization” in treated CNV, called “arteriogenesis,” characterized by the appearance of large-diameter vessels, loss of thin capillaries, and prominent anastomoses of vessels[iv]. Our study reveals flow remodeling occurring with anti-VEGF treatment in neovascular AMD patients. Indeed, the initial CNV pattern observed frequently switched toward a mature pattern after anti-VEGF therapy. Despite the complexity of vascular phenomena involved in neovascular recurrence and remission[v], OCTA suggests that there is a strong correlation between the morphology (and not the size) of CNV and their structural response (or lack thereof) to treatment. OCTA thus highlights the strengths and weaknesses of current therapies, helping to understand the pathophysiological implications of neovascular AMD and to monitor disease progression in treated eyes.

Question №1.       The accuracy of NV-membrane type classification is the most fundamental part of this study. However, only one examiner was responsible for this classification, whereas in most studies at least two examiners conduct classification.

Our team worked with the classification, provided by Coscas et al (2015) for a period of about 3 years, before starting of this study. To reassure our good clinical practice, every six months  we make tests to be sure that there is a concordance between different examiners when assessing an OCT/OCT-A image. Because of this, the examiner in the specific case is one.

Question №2.       In Figure 1, the representative images of each type are shown. However, the quality of the images are poor, especially Figure 1D.

We provide new images with better quality. We apologize for the previous.

Question №3.       In Table 1, all the mean age of males, females, and total patients are the same, 76 years old. Is this a true representation?

It’s not a mistake. To be precise, we put the value with a decimal point to be clearer, that it’s not a mistake.

Question №4.       There are several sentences that need to be referenced.

We put the references where needed. Hope, there’s nothing missing this time

Question №5.       There are several typos. Please edit the manuscript carefully.

Yes. Our deepest apologizes.

Thank you for your precious time and patience.

For any queries, I remain on your disposal.

Best regards,

Dr Kirkova

[i]  Yannuzzi LA, Negrão S, Iida T, et al. Retinal angiomatous proliferation in age-related macular degeneration. Retina 2001;21:416–434.

[ii] Muakkassa NW, Chin AT, de Carlo T, et al. Characterizing the effect of anti-vascular endothelial growth factor therapy on treatment-naive choroidal neovascularization using optical coherence tomography angiography. Retina 2015;35:2252–2259.

[iii] Lumbroso B, Rispoli M, Savastano MC, et al. Optical coherence tomography angiography study of choroidal neovascularization early response after treatment. Dev Ophthalmol 2016; 56:77–85.

[iv] Spaide RF. Optical coherence tomography angiography signs of vascular abnormalization with antiangiogenic therapy for choroidal neovascularization. Am J Ophthalmol 2015; 160:6–16.

[v] Spaide RF. Choroidal neovascularization. Retina 2017;37: 609–610.

Round 2

Reviewer 2 Report

The authors have satisfactorily addressed most of my concerns.